# Rational Design of a Low-Data Regime of Pyrrole Antioxidants for Radical Scavenging Activities Using Quantum Chemical Descriptors and QSAR with the GA-MLR and ANN Concepts

**DOI:** 10.3390/molecules28041596

**Published:** 2023-02-07

**Authors:** Wanting Xie, Sopon Wiriyarattanakul, Thanyada Rungrotmongkol, Liyi Shi, Amphawan Wiriyarattanakul, Phornphimon Maitarad

**Affiliations:** 1Research Center of Nano Science and Technology, College of Sciences, Shanghai University, Shanghai 200444, China; 2Program in Computer Science, Faculty of Science and Technology, Uttaradit Rajabhat University, Uttaradit 53000, Thailand; 3Center of Excellence in Structural and Computational Biology, Department of Biochemistry, Chulalongkorn University, Bangkok 10330, Thailand; 4Program in Bioinformatics and Computational Biology, Graduate School, Chulalongkorn University, Bangkok 10330, Thailand; 5Emerging Industries Institute, Shanghai University, Jiaxing 314006, China; 6Program in Chemistry, Faculty of Science and Technology, Uttaradit Rajabhat University, Uttaradit 53000, Thailand

**Keywords:** QSAR-GA-MLR, QSAR-ANN, pyrrole, radical scavenging activities, antioxidants

## Abstract

A series of pyrrole derivatives and their antioxidant scavenging activities toward the superoxide anion (O_2_^•−^), hydroxyl radical (^•^OH), and 1,1-diphenyl-2-picryl-hydrazyl (DPPH^•^) served as the training data sets of a quantitative structure–activity relationship (QSAR) study. The steric and electronic descriptors obtained from quantum chemical calculations were related to the three O_2_^•−^, ^•^OH, and DPPH^•^ scavenging activities using the genetic algorithm combined with multiple linear regression (GA-MLR) and artificial neural networks (ANNs). The GA-MLR models resulted in good statistical values; the coefficient of determination (*R*^2^) of the training set was greater than 0.8, and the root mean square error (*RMSE*) of the test set was in the range of 0.3 to 0.6. The main molecular descriptors that play an important role in the three types of antioxidant activities are the bond length, HOMO energy, polarizability, and AlogP. In the QSAR-ANN models, a good *R*^2^ value above 0.9 was obtained, and the *RMSE* of the test set falls in a similar range to that of the GA-MLR models. Therefore, both the QSAR GA-MLR and QSAR-ANN models were used to predict the newly designed pyrrole derivatives, which were developed based on their starting reagents in the synthetic process.

## 1. Introduction

Free radicals in organisms can be defined as unstable and highly reactive groups or molecules with unpaired electrons that are constantly produced through intracellular metabolism [1]. They are harmful to the human body, not only aggravating the aging process but also causing a multitude of diseases, including Parkinson’s disease, Alzheimer’s disease, Huntington’s disease, depression, cardiovascular disease, cancers, etc. [2,3,4,5]. In addition to their spontaneous production in an organism, free radicals can emerge abruptly due to several exogenous factors, such as exposure to UV light, alcohol addiction, and excessive smoking [6]. Under physiological conditions, these free radicals usually include oxidizing substances, such as hydroxyl and superoxide anion radicals, hydrogen peroxide, singlet oxygen, nitric oxide, and nitroso peroxide. Many of the conditions caused by these radicals can be prevented by effective antioxidant mechanisms to regulate their presence in the human body [7,8]. Thus, introducing an efficient antioxidant supplement into the body is a promising solution.

A vast number of food-derived compounds have been identified as natural antioxidants, such as tocopherols, ascorbic acid, vitamin A, coumarin derivatives, flavonoid groups, pyrrole derivatives, and so forth [9,10,11,12,13,14]. Reactive oxygen species (ROS) are generated in living organisms during metabolism in the form of superoxide anions (O_2_^•−^), hydroxyl radicals (^•^OH), hydrogen peroxide (H_2_O_2_), and nitric oxide (NO). Therefore, various in vitro assays exist to measure the ROS-scavenging ability of compounds, such as those determining 1,1-diphenyl-2-picryl-hydrazyl (DPPH^•^), 2,2′-azino-bis(3-ethylbenzothiazoline-6-sulfonic acid) (ABTS^+•^), O_2_^•−^, and ^•^OH scavenging activity; ferric ion (Fe^3+^) and cupric ion (Cu^2+^) reducing power; and ferrous ion (Fe^2+^) and Cu^2+^ chelating activity, compared with the positive controls Trolox or BHT (standard antioxidant compounds) [14,15,16]. Numerous natural compounds are potent antioxidants. However, the extraction and purification of natural bioactive compounds (drugs) is time-consuming and expensive [17] and, hence, not conducive to industrial manufacturing. Thus, one of the top trending research topics in organic chemistry is the rapid and environmentally friendly production of ideal (non-) natural chemicals.

Significant progress has been made in the development of the quantitative structure–activity relationship (QSAR) method in the field of drug discovery. Understanding the roles of this technique is notably useful in interpreting the molecular biological activity and developing new chemical designs [18]. To study the inhibitory effects of 12 wine-extracted flavonoids on lipid peroxidation, QSAR models have been developed via polynomial and multiple regression using lipophilicity and molar refractivity descriptors [19]. Further, the partial least square (PLS) regression and artificial neural networks (ANNs) approaches have been used to investigate the relationship between the biological activities of 35 isonicotinamide derivatives (protein kinase inhibitors of GSK-3β) and their molecular descriptors. The generated robust models with *R*^2^ values greater than 0.8 have been considered promising for predicting new highly active molecules for the inhibition of Alzheimer’s disease [20]. Recently, with the development of complex computing architectures, QSAR models used for pharmacological research have been incorporated into some computing tools based on machine learning and deep learning algorithms, which have shown attractive and stable performance [21,22].

QSAR models relying on simple physicochemical descriptors have been described in early studies but are insufficient for generating comprehensive structure–activity relationships [23]. The electronic descriptors obtained from quantum chemical calculations play a more significant role in QSAR models, especially concerning antioxidant compounds [24,25,26]. In recent years, some comparative QSAR studies have shown that employing the descriptors generated using quantum chemical methods can improve the accuracy of the results and lead to more reliable QSAR models [27,28,29]. Therefore, in this work, the QSAR technique was applied to derive a mathematical model relating the structural properties of pyrrole derivatives with their antioxidant activities. Three types of antioxidant activities were measured, namely, those against DPPH^•^, O_2_^•−^, and ^•^OH. Ab initio quantum chemical calculations were employed to calculate the steric and electronic molecular descriptors since the QSAR analysis was carried out on a small data set. From the many molecular descriptors, the genetic algorithm (GA) was used to select descriptors that resulted in the best fit to models using multiple linear regression (MLR). The interpretability, clarity, and understandability of the models presented by using MLR make it a suitable choice for modeling. At the same time, the complex relationship between the molecular descriptors and their antioxidant activities provides a very good justification for using the ANN-based nonlinear method. The various QSAR models could be beneficial in predicting the ^•^OH, O_2_^•−^, and DPPH^•^ radical scavenging activities of the newly designed pyrrole compounds, which also provide their starting chemical compounds for the synthetic route.

## 2. Results and Discussion

### 2.1. Scavenging Activities of Studied Pyrrole Derivatives

Figure 1 depicts the three ROS scavenging activities of the 15 pyrrole derivatives, where the percentage scavenging activities toward ^•^OH and O_2_^•−^ are relatively close, indicating a similar range of ROS reactivities. The values for ^•^OH scavenging occur in the range of 6.365 to 9.151, and **Cpd.1**, **2**, **3**, **12**, **13**, **14**, and **15** show more than 80% ^•^OH scavenging activity. The values for the O_2_^•−^ antioxidants are in the range of 6.203 to 8.644, with **Cpd.3** and **12** showing high O_2_^•−^ antioxidant activity. Regarding DPPH^•^ scavenging, **Cpd.11** has the lowest value, at 13.48%, while the highest activity is 76.04% for **Cpd.7**. It is worth noting that **Cpd.2**, **7**, **12**, **13**, and **15** exhibit good scavenging activities for all the ROS types. More details about the pyrrole derivatives see at the methods section. Subsequently, to investigate the relationship between the ROS scavenging activities and the structural properties of the pyrrole derivatives, the QSAR mathematical models were applied, and both linear and non-linear models were developed for all three free radicals scavenging activities.

### 2.2. QSAR GA-MLR Models

Before carrying out the QSAR modeling, we first calculated the correlation coefficients of variable pairs, based on the preset definitions of the antioxidant activities as the dependent variables and the molecular descriptors as the independent variables. The values of the pairwise correlation coefficient, *r*, lie between 1 and −1; the correlation coefficients are all presented in the correlation matrix heatmap (Figure 2). Some of the descriptors show similar importance concerning Y1 and Y2; the most significant is the C4–C11 bond length, and its coefficients with Y1 and Y2 are −0.73 and −0.71, respectively. This can be explained by noting that the shorter C4–C11 bond length leads to better antioxidant activity in scavenging ^•^OH and O_2_^•−^ free radicals. The charges of O8 and O12 are the next two important features, both of which show negative effects on antioxidant activities, with the absolute value of the correlation coefficients exceeding 0.5. Therefore, more negative charges on these two oxygen atoms helps improve the ROS scavenging efficiency of the pyrrole derivatives.

Examining the correlation coefficients between the descriptors and the DPPH^•^ scavenging activity, Y3, shows that the N1-C13 bond length is highly correlated with the latter, with a value of 0.75, but has little effect on the other two types of free radical scavenging activities. AlogP, which reflects molecular hydrophobicity, is the second important descriptor for improving Y3 performance, with an *r* value of −0.49. Thus, hydrophilicity and longer distances between the R(d) substituent and pyrrole ring of the antioxidants are preferred in achieving a higher DPPH^•^ quenching ability. Table 1 presents the QSAR model results using the GA-MLR method.

QSAR GA-MLR of ^•^OH model. Equation (1) (see Table 1) is the QSAR model obtained from the GA-MLR method for ^•^OH scavenging activity. The model yields good statistical values, with *R*^2^ = 0.848 and *R*^2^*(CV)* = 0.711. The regression is significant since *F* > *F_cr_*. Equation (1), Y1 (^•^OH scavenging activity) = −90.879 ∗ X17 (bond C2-R(b)) − 47.988 ∗ X19 (bond C4-C11) + 0.016 ∗ X20 (polarizability) + 207.384, implies that the steric structural properties (C2-R(b) and C4-C11 bonds) play an important role in the ^•^OH scavenging activity, while the electronic polarizability is a minor property. Further, the negative coefficient values of X17 and X19 suggest that the shorter bond distances of C2-R(b) and C4-C11 are favorable in increasing the ^•^OH scavenging activity.

QSAR GA-MLR of O_2_^•−^ model. The QSAR model obtained from the GA-MLR method for O_2_^•−^ scavenging activity is given in Equation (2) (Table 1), with *R*^2^ = 0.863 and *R*^2^*(CV)* = 0.731. The linear regression model is significant since *F* > *F_cr_*. Considering Equation (2), Y2 (O_2_^•−^ scavenging activity) = −43.836 ∗ X19 (bond C4-C11) + 0.005 ∗ X20 (polarizability) − 75.277 ∗ X21 (HOMO energy) + 47.527, both electronic (HOMO energy) and steric properties are mainly related to the O_2_^•−^ scavenging activity. In Equation (2), X19 has a negative coefficient, implying that the shorter bond distance of C4-C11 (see the position in Figure 5) is favorable in increasing the O_2_^•−^ scavenging activity, similar to the case in Equation (1) for the ^•^OH scavenging activity. Regarding the HOMO energy, the pyrrole derivatives with higher values are preferable in increasing the O_2_^•−^ scavenging activity. Further, Y1 (^•^OH scavenging activity) and Y2 (O_2_^•−^ scavenging activity) have a high Pearson correlation coefficient of 0.84; from the molecular properties related to these ROS, X19 and X20 are common to both models.

QSAR GA-MLR of DPPH^•^ model. The QSAR model for DPPH^•^ scavenging activity developed using the GA-MLR method is given in Equation (3) (see Table 1). The statistical results include *R*^2^ and *R*^2^*(CV)* values of 0.810 and 0.559, respectively, which are slightly lower than those of the ^•^OH and O_2_^•−^ scavenging activity models. The DPPH^•^ scavenging activity (Y3) is related to the N1-C13 bond, AlogP, and the Connolly surface area, as shown in Equation (3): Y3 = 61.220 ∗ X16 (N1-C13 bond) − 1.240 ∗ X26 (AlogP) + 0.052 ∗ X30 (Connolly surface area) − 102.072. The bond distance of N1-C13 (see the position in Figure 1) has the highest (positive) coefficient value, suggesting that a longer bond distance would support the DPPH^•^ scavenging activity, which corresponds well with experimental discussions [30]. The second notable feature is that molecules with lower AlogP values are preferred, that is, antioxidants need to be more hydrophilic to have a higher quenching ability toward radicals. In addition, the compounds with higher values of Connolly surface area are beneficial to improve the DPPH^•^ scavenging activity; however, this property only plays a minor role in the DPPH^•^ scavenging activity.

The molecular descriptors used in Equations (1)–(3) of the QSAR GA-MLR models consist of steric and electronic properties (Table 2). The important descriptors from the QSAR models correspond well with the Pearson correlation coefficients of the ^•^OH, O_2_^•−^**,** and DPPH^•^ scavenging activities. Figure 3a–c depict the linear relationship between the experimental and predicted ^•^OH, O_2_^•−^**,** and DPPH^•^ scavenging activities, and the predicted results are listed in Appendix A. Based on the residuals between the predicted and actual activities, the *RMSE* of the training set (*RMSE* train) of the ^•^OH, O_2_^•−^**,** and DPPH^•^ scavenging activities were calculated as 0.368, 0.269, and 0.958, respectively, as shown in Figure 3a–c. To evaluate the feasibility of the QSAR models, the test set (**Cpd.6**, **8**, and **11**) predicted the compounds’ ^•^OH, O_2_^•−^**,** and DPPH^•^ scavenging activities using Equations (1)–(3), respectively. The *RMSE* values of the test set (*RMSE* test) of the ^•^OH and DPPH^•^ scavenging activities (Figure 3a,c) are lower than the *RMSE* train values, while the *RMSE* test for O_2_^•−^ scavenging is slightly higher than the *RMSE* train (Figure 3b). Therefore, in summary, the QSAR GA-MLR technique helps manipulate mathematical models for limited data sets. Furthermore, the obtained models could be used for further predictions of newly designed pyrrole compounds to demonstrate their good predictive power in external evaluations. 

### 2.3. QSAR ANN Models

The ANN models originate from Artificial Intelligence, which is an interconnected assembly of simple processing elements, known as artificial neurons, that mimic human neuron functions. Consequently, the input of each neuron is one or more weighted variables, and the output is a linear or nonlinear function of the weighted inputs. Alternatively, the neurons learn by adjusting the weights of the input variables by minimizing the error between the neuron’s expected output and the measured output value. Therefore, we applied the QSAR-ANN technique for the ^•^OH, O_2_^•−^**,** and DPPH^•^ scavenging activities and the selected molecular descriptors (Table 2). The ANN architecture was set as 3-3-1, with one input layer (three neurons), one hidden layer (three neurons), and one output layer, according to the descriptors found in Equations (1)–(3) for the ^•^OH, O_2_^•−^**,** and DPPH^•^ scavenging activities, respectively. The optimal ANN models of the three ROS are given in Table 3. The statistical *R*^2^ values of the ANN models are in the range of 0.920–0.965, which is higher than that of the GA-MLR models (*R*^2^ values in the range of 0.810–0.863). The linear plots of the experimental versus predicted ^•^OH, O_2_^•−^**,** and DPPH^•^ scavenging activities from the ANN models are displayed in Figure 3d–f, respectively. The *RMSE* train values of all ANN models fall in the range of 0.175–0.427, which is much lower than that of the GA-MLR models. However, the *RMSE* test values of the ANN models are comparable to the results from the GA-MLR ones.

In summary, as shown in Table 3, the ANN learns to predict the antioxidant activity with higher accuracy, approximating the experimental data with small differences (see the predicted data in Appendix A). Therefore, the three ANN models can be used to predict the antioxidant activity of the newly designed pyrrole compounds.

### 2.4. Newly Designed Compounds with Predicted ROS Activities

Figure 1 provides insight into the newly designed pyrrole antioxidants. Firstly, 4-hydroxycoumarin (1a) and 2-hydroxy-1,4-naphthoquinone (1b) were selected for substitution at the R(a) position since they appeared in **Cpd.2**, **12**, and **15**, which gave superior ROS activities. Secondly, at the R(b) substituent position, a thiophene ring was chosen as it resulted in shortening the C2-R(b) bond distance, as observed in the X17 property of **Cpd.3**, **8**, and **15** (Appendix A). Next, in the R(c) position, the substitution of cyclohexanone or benzoyl groups has a prevailing impact on the scavenging activity, as seen in **Cpd.1**, **2**, **12**, and **14**. In addition, this position is related to X19 (bond C4-C11); the shorter the bond distance, the more preferable the antioxidant activity. Lastly, regarding the R(d) position, related to the X16 property (bond N1-C13), a longer N1-C13 bond distance would support improved antioxidant activity. Thus, three functional groups—4-methoxyphenyl, n-butylamine, and phenylethylamine—were selected for substitution at the R(d) position. In summary, there are 17 new compounds, **Cpd.16** to **Cpd.32** (Figure 1). All new complex structures were built and optimized by employing the same computational level criteria as those for the training set compounds (see at methods section). The structural and electronic properties were then collected and are presented in Appendix A for the predicted ^•^OH, O_2_^•−^**,** and DPPH^•^ scavenging activities of all newly designed molecules.

The ^•^OH, O_2_^•−^, and DPPH^•^ scavenging activities of newly designed pyrrole compounds were then predicted by using both QSAR GA-MLR and QSAR-ANN (see in Appendix A). The plots of number of compounds with their predicted ^•^OH, O_2_^•−^, and DPPH^•^ scavenging activities are depicted in Figure 4. The ^•^OH and O_2_^•−^prediction results obtained from the two models shares a similar trend, while the predicted DPPH^•^ scavenging activities from GA-MLR and ANN showed some partial differences. It is worth noting that there are seventeen, and nine new pyrrole compounds were predicted to achieve more than 80% of ^•^OH and O_2_^•−^ scavenging activities, respectively (Appendix A). For the predicted DPPH^•^ scavenging activities above 70%, there are found on **Cpd.26**, **27**, **31**, and **32** (Figure 6). In addition, these four new compounds have also resulted in great ^•^OH and O_2_^•−^ scavenging activities.

In summary, our newly designed pyrrole compounds (**Cpd.16**–**Cpd.32**) based on the QSAR molecular descriptors showed the higher tendency of ^•^OH, O_2_^•−^, and DPPH^•^ scavenging activities in comparison with training data of pyrrole derivatives (see in Appendix A).

## 3. Methods

Experimental activities data. In the current QSAR study, we employed 15 pyrrole derivatives and obtained their experimental radical scavenging activities (Appendix A) from Tania et al. [30]. Figure 5 depicts the template of a pyrrole ring with four substitution positions. Three types of radical scavenging activities (against ^•^OH, O_2_^•−^, and DPPH^•^) were measured for all the pyrrole derivatives (Figure 6) and used as the data sets in this work. Therefore, we represented the ^•^OH, O_2_^•−^, and DPPH^•^ radical scavenging activities with three dependent variables: Y1, Y2, and Y3, respectively.

**Figure 5 molecules-28-01596-f005:**
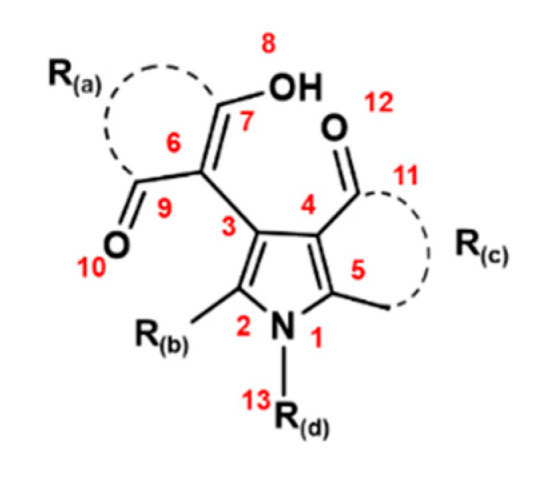
The main skeleton of pyrrole derivatives.

Molecular Features. To obtain the electronic and steric molecular descriptors, all the pyrrole derivatives were built and optimized using the Hartree–Fock (HF) functional and the 6-31G(d,p) basis set, which includes the polarization functions of all atoms in the structure. The optimizations were performed with the Gaussian 16 program [31]; the optimized structures were analyzed, and 23 of their molecular properties, denoted as X1–X23, were recorded (Appendix A). Furthermore, additional molecular properties, denoted as X24–X33 (Appendix A), were obtained using the Materials Studio software [32], leading to a total of 33 independent variables in this work.

**Figure 6 molecules-28-01596-f006:**
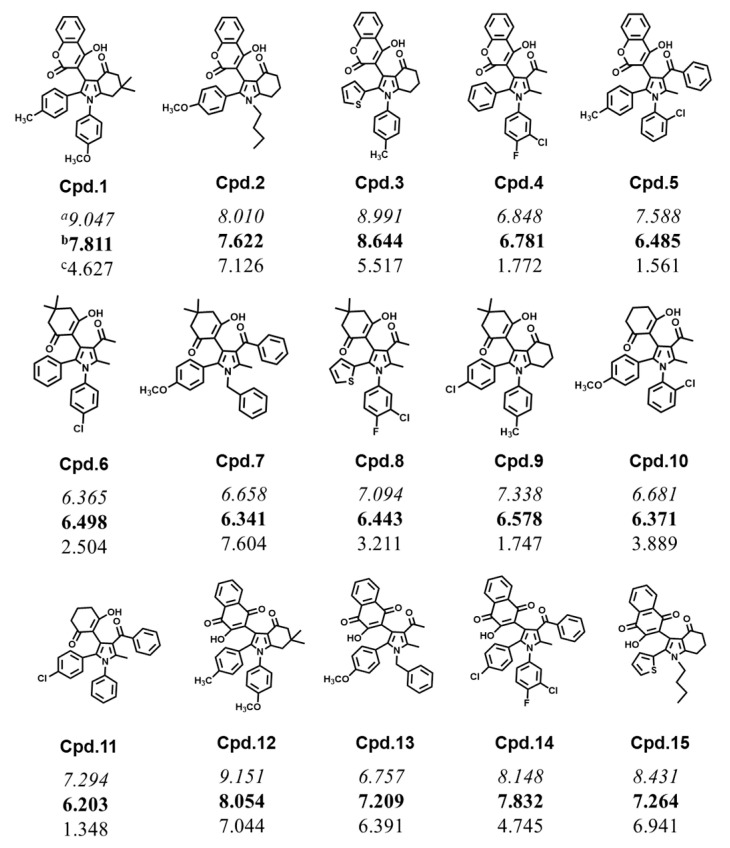
Pyrrole derivatives and their radical scavenging activities. The data are expressed as the (percent scavenging activity)/10 toward ^a •^OH at a compound concentration of 80 μM (in italics), ^b^ O_2_^•−^ at 20 μM (in bold), and ^c^ DPPH^•^ at 90 μM (in normal font) [30].

Data sets. The data of pyrrole derivatives were divided into a training set (80%) and a test set (20%) according to the Kennard–Stone algorithm [33] using the Python package Kennard-stone 1.1.2. Based on this algorithm, **Cpd.6**, **8**, and **11** were selected as the test-set compounds, that is, for use as the external test set to evaluate the generalization performance of the regression models.

GA-MLR method. Inspired by natural genetics and evolution, genetic function approximation (GFA) is an approach that emphasizes achieving model-building optimization. The GFA method has been used in the development of QSAR models and has demonstrated the ability to elucidate the relationship between the desired molecular activity and chemical identity [34,35]. It automatically selects variables and effectively discovers combinations of features that take advantage of correlations between multiple features. The maximum number of variables is established by fixing the preferred model length. Additionally, the GFA algorithm can work flexibly with or without spline curves [36], which increases the complexity of the model, though at the expense of reducing its interpretation ability. The expression of the output equation without splines is the same as that of the MLR model. Therefore, the MLR model based on the GFA algorithm (GA-MLR) can provide an “understanding” of important molecular characteristics for the activity of compounds. One notable feature of the GFA is that it can generate a set of models, rather than a single model, at once. The workflow of the GA can be summarized as a basic function of genetic selection. After crossover and mutation operations, new generations will be generated. Each new model is then scored according to a specific fitness criterion.

The regression analysis was developed using the GFA module in Materials Studio. Initially, the training data was fully imported, with the maximum equation length set at 3 and the population and maximum generations set to 1000 and 500, respectively. The mutation probability was 0.1. The fitness of a GFA model was measured using the R-squared (*R*^2^) value, which reflects the fraction of the total variance of the dependent variable, *y*; the larger the *R*^2^ value, the better the model.

ANN method. ANN is a nonlinear-function mapping technique originally developed to simulate the structure and computations of the brain. In the field of cheminformatics, it has been widely used to study the complex nonlinear relationship between the biological activity of molecules and their structural characteristics [37,38,39,40]. In this study, one of the most popular neural networks, the multilayer perceptron (MLP) ANN, which served as the function approximation method, was used to model the antioxidant activities and structural properties data [41]. The MLP network designed herein is based on the principle of the backpropagation algorithm and was optimized using the Levenberg–Marquardt technique to reduce the error [42]. Generally, the MLP network includes three types of neural layers: an input layer, one or more hidden layers, and an output layer.

The running script was generated using the neural network fitting tool in the MATLAB program [43], and a multilayer ANN structure composed of three input neurons, an implicit layer (three neurons), and an output layer (one neuron) was constructed. In our ANN regression task, the Bayesian regularization backpropagation algorithm was used to optimize and update the weights and biases, which are the network optimization functions according to the Levenberg–Marquardt algorithm. The optimal combination was determined by minimizing the combination of the square error and the weight to generate a network model with a good generalization ability. This process is also known as Bayesian regularization.

Evaluation of statistical terms. This section discusses the equations used to evaluate the prediction reliability of the QSAR models. The coefficient of determination (*R*^2^) measures how well a statistical model predicts an outcome. It is the proportion of variance in the dependent variable that is explained through the model; the closer the value is to 1.0, the better the genetic function approximation equation explains the dependent variable. The expression for *R*^2^ is given using Equation (1):(1)$$R^{2}=\frac{ESS}{TSS}\;$$
where *ESS* is the sum of squares of errors (or the explained sum of squares), and *TSS* is the total sum of squares of *y*. The variation in *y* not explained through the regression equation (or the residual sum of squares, *RSS*) is the sum of the squares of the differences between the predicted values (${\overset{´}{y}}_{i}$) and the actual ($y_{i}$) as given in Equation (2):(2)$$RSS=\sum_{i=1}^{n}{\left({\overset{´}{y}}_{i}-y_{i}\right)}^{2}$$

The total variation in *y* (or the total sum of squares, *TSS*) is the sum of the squares of the differences between the observed *y* values ($y_{i}$) and their mean ($\bar{y}$). It can also be described as the mean-corrected sum of squares of the responses over the entire data set. The *TSS* is given as in as in Equation (3):(3)$$TSS=\sum_{i=1}^{n}{\left(y_{i}-\bar{y}\right)}^{2}$$

The *TSS* is also expressed as in Equation (4):(4)TSS=ESS+RS

The variation in *y* explained through the regression equation (*ESS*) is the sum of the squares of the differences between the predicted *y* values $\left({\overset{´}{y}}_{i}\right)$ and the mean $\left(\bar{y}\right)$, as given in Equation (5):(5)$$ESS=\sum_{i=1}^{n}{\left({\overset{´}{y}}_{i}-\bar{y}\right)}^{2}$$

The F test is a standard statistical test to assess the equality of the variances of two populations with normal distributions. Here, it was used to test whether the variance in the data that is explained through the regression is significantly larger than the remaining variance due to errors. If this is the case, the model is then stated to be significant rather than one that simply fits the noise. The significance-of-regression (SOR) *F* value is defined in Equation (6):(6)$$F=\frac{RSS/\left(p-1\right)}{ESS/\left(n-p\right)}$$
where *n* is the number of data points from which the model is built, and *p* is the number of parameters in a regression model (including the intercept, when present).

The calculated *F* value was compared with the tabulated values of the *F* distribution for different values of *n* and *p*. The critical SOR *F* value is the critical point of the F distribution of degrees *n* − *p* and *p* − 1 evaluated for a probability of 0.05 (at a 95% confidence level). The regression is significant if *F* is greater than the tabulated value *F_cr_*, or SOR *F* value (95%).

The cross-validation *R*^2^, or *R*^2^*(CV)*, is the cross-validated equivalent of *R*^2^, which constitutes a crucial measure of a model’s predictive power; the closer the value is to 1.0, the better the predictive power. For a good model, *R*^2^*(CV)* should be reasonably close to *R*^2^. *R*^2^*(CV)* is expressed as in Equation (7):(7)$$R^{2}\left(CV\right)=1-\frac{PRESS}{TSS}$$

The cross-validation involved excluding the required set of data, performing the principal component analysis (PCA) on the remaining data, and calculating the *PRESS* of the prediction error based on the model generated using the retained data, which was excluded from model development. This process was repeated until each observation was ignored. The *PRESS* is calculated as in Equation (8):(8)$$PRESS\left(n\right)=\sum_{i=1}^{n}{\left(y_{i}-{\overset{´}{y}}_{i}\right)}^{2}$$

The root mean square error (*RMSE*) is used to determine whether a model has the predictive ability, reflected using *R*^2^, to ensure its rationality from a statistical perspective. The *RMSE* is the square root of the sum of the squared differences between the actual and predicted values divided by the number of observations, *N*, as given in Equation (9). It measures deviations from true values and is sensitive to divergent data.
(9)$$RMSE=\sqrt{\frac{1}{N}\overset{n}{\underset{i=1}{\sum }}{\left(y_{i}-{\overset{´}{y}}_{i}\right)}^{2}}$$

## 4. Conclusions

The QSAR concept was applied to understand the influence of substitutions on pyrrole derivatives and their ^•^OH, O_2_^•−^, and DPPH^•^ scavenging activities. Both the GA-MLR and ANN techniques were applied to relate the quantitative relationships between the three types of antioxidant activities of the pyrrole derivatives and their molecular descriptors, which were determined from quantum chemical calculations. In the QSAR GA-MLR models, the statistical coefficient of determination, *R*^2^, was greater than 0.8, while the QSAR-ANN models yielded superior *R*^2^ values (greater than 0.9), both of which showed high predictive ability. The *RMSE* of the test set was introduced to evaluate the prediction reliability of all QSAR models; the *RMSE* values were in the range of 0.3–0.6, which implies that the models can be used for further predictions. However, the *RMSE* of the ANN models test set did not outperform substantially in comparison with the GA-MLR model. Thus, in this case, the resulted GA-MLR model have equivalent prediction reliability with the ANN model. The obtained QSAR GA-MLR models, both steric (bond lengths and Connolly surface area) and electrostatic (HOMO energy and polarizability) properties, played an important role in the three types of antioxidant activities equations. Finally, based on the QSAR GA-MLR and QSAR-ANN models, most of the predictions for the ^•^OH, O_2_^•−^, and DPPH^•^ scavenging activities of the newly designed pyrrole compounds were more effective than those of the training set pyrrole derivatives. Based on our findings, the newly designed compounds **Cpd.26**, **27**, **31**, and **32** were predicted via both the GA-MLR and ANN models to be potent and effective antioxidants against ^•^OH, O_2_^•−^, and DPPH^•^, which would be useful in further experimental syntheses and tests.

## Data Availability

Not applicable.

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
