# Peer review of "Rational Design of a Low-Data Regime of Pyrrole Antioxidants for Radical Scavenging Activities Using Quantum Chemical Descriptors and QSAR with the GA-MLR and ANN Concepts"

_molecules, 2023, doi:10.3390/molecules28041596_

Round 1
Reviewer 1 Report (Previous Reviewer 1)
The revised version of the manuscript entitled “Rational design on low-data regime of pyrrole antioxidants for radical scavenging activities using quantum chemical descriptors and QSAR with GA-MLR and ANN concepts” by Wanting Xie et al. describes the application of fuzzy logic methods (genetic algorithms and artificial neural networks) for modeling the activity-structure relationships of antioxidant pyrrole derivatives.
Generally, the quality of the paper has been improved considerably – Authors addressed correctly the issues, that I mentioned in my 2 previous reviews. In the current version I noticed 2 mistakes.
1. Line 310. It reads ‘Figures 5a-c depicts’, it should read ‘Figures 5a-c depict’
2. In References section Authors use sometimes full names of journals, for instance, Ref. 9, 11, 24, etc. Please, use journal abbreviations.
Author Response
Reviewer response: Thank you for your caring and kind review of this work. All comments above were edited according to the reviewer’s suggestions.
Reviewer 2 Report (Previous Reviewer 2)
Dear Author,
This work you have prepared is well organized in every aspect. However, I think that it will be more effective and valuable for the study to deal with the conclusion part in more detail.
Author Response
Reviewer response: Thank you for your kind support. The conclusion part was revised as shown in line 397-408.
This manuscript is a resubmission of an earlier submission. The following is a list of the peer review reports and author responses from that submission.
Round 1
Reviewer 1 Report
The submitted manuscript entitled “Rational design on low-data regime of pyrrole antioxidants for radical scavenging activities using quantum chemical descriptors and QSAR with GA-MLR and ANN concepts” by Wanting Xie et al. describes the application of fuzzy logic methods (genetic algorithms and artificial neural networks) for modeling the activity-structure relationships of antioxidant pyrrole derivatives.
On the whole, the reviewed paper is within the scope of the Molecule journal and the obtained findings seem to be appealing to the scientific community. On the other hand, many major issues prevent the text from suggesting for publication in Molecules.
1. First of all, the extensive editing of English language is urgently needed, because many grammar, spelling and punctuation errors appeared in the text, that is definitely not allowed in the scientific text, for instance:
line 8, it reads ‘CHINA’, it should read ‘China’
abstract: the word repetition should be avoided ‘this work’
line 30, it reads ‘to predict the newly designed of pyrrole derivatives’, it should read ‘to predict the newly designed activities of pyrrole derivatives’
line 31, it reads ‘which designing based on’, it should read ‘which designing was based on’
line 43, ‘Most of the pathetic’?
line 72, it reads ‘relationships.28’, it should read ‘relationships [28].’
line 72, it reads ‘descriptor’, it should read ‘descirptors’
line 74, it reads ‘descriptors has’ it should read ‘descriptors have’
line 74, what does mean the vague statement, that ‘descriptors has (originally written) some potential’
line 78, it is not clear, how QSAR technique was applied to ‘manipulate the mathematic model’? What does it mean?
line 89, dot is missing at the end of the sentence.
line 101, ‘Hartree-Fock method’ not ‘Fork’
page 6, it reads ‘y’ it should read ‘Y’
page 6, in this context ‘tested compounds’ are ‘training subset’, and ‘untested’ are ‘test subset’. It should be unified in the text.
page 6, what is ‘PLS dimension’. Is it a model complexity (number of latent variables)? It should be clarified.
and many, many other mistakes.
Authors should carefully check and correct the whole text.
2. Table 1 is not informative; therefore it might be moved to Supplementary Materials. It is generally accepted in chemoinformatics/chemometrics that molecules are placed in rows and properties (descriptors in columns); therefore it should be changed. It is not necessary to name molecules ‘Cpd.12’. Better is ‘12’.
3. Authors stated that GA produces better models than stepwise regression methods. Please, compare the GA and ANN selection method with stepwise procedures or UVE-PLS approach to justify the above statement.
4. Why only the random manner of training/test subset selection was applied in the study? Please, compare the obtained finding using Kennard-Stone’a or Duplex procedures.
5. Authors use word ‘excellent ability’ in the text. What does it mean?
6. In Figure 6 the set of descriptors selected by the GA and ANN vary considerably for different ROS. Please, provide any explanation of the fact.
7. The validity and the predictive power of the models should be checked carefully. The cross-validation procedure is not sufficient. Please, check whether the provided models fulfill the Golbraikh-Tropsha criterion (‘Beware of q2’).
8. For each three antioxidants 11 ANN models are obtained with a completely different net architecture (number of neurons and layers). Please, provide the explanation of the fact.
9. Have the Authors compared the predicted activities for the newly designed molecules?
I strongly encourage the Authors to rearrange the text of the manuscript, because it is vaguely written that makes it difficult to understand it properly.
Author Response
Dear reviewer,
Thank you so much for your great suggestions and comments. We have revised the manuscript and replied to the questions. Please see our response in the attached file.
Best regards.

Reviewer 2 Report
Dear Authors,
The study is very comprehensive study.
1. Which properties or conditions of the pyrrole derivatives selected in the study were chosen?
2. Why was only HF/6-31G(d,p) chosen as the basis for the determination of the quantum chemical computational parameters of the study, or were other sets unthinkable? It will be helpful in the study if you explain.
3. Why were steric parameters not taken into account besides the structural and electronic properties? Doesn't organizing the study by considering this parameter ensures the validity of the study and its compatibility with reality?
4. Increasing the resolution of Figure 5 will be helpful in evaluating the results.
5. The QSAR study was organized using two models (QSAR GA-MLR and QSAR ANN). The reason why both methods were chosen and why the results produced similar trends should be examined both in the evaluation part and in the conclusion part.
Regards,
Author Response
Dear Reviewer,
Thank you so much for your great suggestions and comments. The revised manuscript is improved, and we edited it according to your suggestions.
All comments were replied to one by one as seen in the attached file.
Best regards.

Round 2
Reviewer 1 Report
The revised version of the manuscript entitled “Rational design on low-data regime of pyrrole antioxidants for radical scavenging activities using quantum chemical descriptors and QSAR with GA-MLR and ANN concepts” by Wanting Xie et al. describes the application of fuzzy logic methods (genetic algorithms and artificial neural networks) for modeling the activity-structure relationships of antioxidant pyrrole derivatives.
The Authors addressed partially the issues mentioned in my previous review; however there are still major issues in the revised text.
I strongly recommend the extensive editing of English language by a native speaker or a specialized agency, because it is really hard to understand it properly. Still, a lot of grammar, spelling and punctuation errors appear in the text. Let me show just a few most noticeable, that are not allowed in the scientific text, for instance:
line 36, it reads ‘will not only aggravates’, it should read ‘will not only aggravate’
line 78, it reads ‘playing’, it should read ‘play’
line 100, it reads ‘which collecting’, it should read ‘which were collected’
line 109, ‘Hartree-Fock method’ not ‘Fork’
line 110, it reads ‘optimization were’, it should read ‘optimization was’
line 170, it reads ‘To evaluated’, it should read ‘To evaluate’
line 167, the word repetition should be avoided in one sentence (‘combination’)
line 350, it reads ‘lines shows’, it should read ‘lines show’
line 364, word repetitions ‘as seen from seen from’
and many, many others.
2. Lines 63-73 are not informative; therefore not necessary.
3. Line 88, It is contradictory statement ‘carried out on a small data set. With many molecular descriptors’
4. The description of data sets is repeated a few times in the text. ‘Y’ or ‘y’? It should be unified.
5. The values in Figure 4 are repeated. The redundant data should be reduced; therefore the triangular matrix is sufficient.
6. The reference list must be checked, corrected and unified.